# Feeding a Negative Dietary Cation-Anion Difference to Female Goats Is Feasible, as Indicated by the Non-Deleterious Effect on Rumen Fermentation and Rumen Microbial Population and Increased Plasma Calcium Level

**DOI:** 10.3390/ani11030664

**Published:** 2021-03-02

**Authors:** Kang Yang, Xingzhou Tian, Zhengfa Ma, Wenxuan Wu

**Affiliations:** 1Institute of Animal Nutrition and Feed Science, College of Animal Sciences, Guizhou University, Guiyang 550025, China; 13639021201@163.com (K.Y.); tianxingzhou@yeah.net (X.T.); m15117471541@163.com (Z.M.); 2Institute of New Rural Development, Guizhou University, Guiyang 550025, China

**Keywords:** dietary cation-anion difference, rumen fermentation, rumen microbial population, plasma calcium, female goat

## Abstract

**Simple Summary:**

Diets with a lower dietary cation-anion difference could prevent hypocalcemia, enhance health, and extend the economic life of transition mammary animals. However, there is less information on rumen fermentation, cellulolytic bacteria populations, and microbiota for female goats fed a negative dietary cation-anion difference diet. We speculate that a negative dietary cation-anion difference would not affect the rumen fermentation parameters. Therefore, the present study was conducted to evaluate the effect of a negative dietary cation-anion difference diet on rumen pH, buffering capability, volatile fatty acids of acetic acid, propionic acid, butyric acid, total volatile fatty acid and acetic acid/propionic acid profiles, ruminal cellulolytic bacteria populations, and microbiota. These results provide a further evaluation on the feasibility of feeding a negative dietary cation-anion difference diet to goats.

**Abstract:**

The dietary cation-anion difference (DCAD) has been receiving increased attention in recent years; however, information on rumen fermentation, cellulolytic bacteria populations, and microbiota of female goats fed a negative DCAD diet is less. This study aimed to evaluate the feasibility of feeding a negative DCAD diet for goats with emphasis on rumen fermentation parameters, cellulolytic bacteria populations, and microbiota. Eighteen female goats were randomly blocked to 3 treatments of 6 replicates with 1 goat per replicate. Animals were fed diets with varying DCAD levels at +338 (high DCAD; HD), +152 (control; CON), and −181 (low DCAD; LD). This study lasted 45 days with a 30-d adaption and 15-d trial period. The results showed that the different DCAD levels did not affect the rumen fermentation parameters including pH, buffering capability, acetic acid, propionic acid, butyric acid, sum of acetic acid, propionic acid, and butyric acid, or the ratio of acetic acid/propionic acid (*p* > 0.05). The 4 main ruminal cellulolytic bacteria populations containing *Fibrobacter succinogenes*, *Ruminococcus flavefaciens*, *Butyrivibrio fibrisolvens*, and *Ruminococcus albus* did not differ from DCAD treatments (*p* > 0.05). There was no difference in bacterial richness and diversity indicated by the indices Chao, Abundance Coverage-based Estimator (Ace), or Simpson and Shannon, respectively (*p* > 0.05), among 3 DCAD levels. Both principal coordinate analysis (PCoA) weighted UniFrac distance and unweighted UniFrac distance showed no difference in the composition of rumen microbiota for CON, HD, and LD (*p* > 0.05). At the phylum level, Bacteroidetes was the predominant phylum followed by Firmicutes, Synergistetes, Proteobacteria, Spirochaetae, and Tenericutes, and they showed no difference (*p* > 0.05) in relative abundances except for Firmicutes, which was higher in HD and LD compared to CON (*p* < 0.05). At the genus level, the relative abundances of 11 genera were not affected by DCAD treatments (*p* > 0.05). The level of DCAD had no effect (*p* > 0.05) on growth performance (*p* > 0.05). Urine pH in LD was lower than HD and CON (*p* < 0.05). Goats fed LD had higher plasma calcium over HD and CON (*p* < 0.05). In summary, we conclude that feeding a negative DCAD has no deleterious effects on rumen fermentation and rumen microbiota and can increase the blood calcium level, and is therefore feasible for female goats.

## 1. Introduction

Dietary cation-anion difference (DCAD; mmol/kg dry matter (DM)) refers to the difference between the number of millimoles of major cations (Na^+^ and K^+^) and major anions (Cl^−^ and S^2−^) per kg DM in the diet [1]. The initial benefit of feeding a negative DCAD was reported by Block [2], who observed that treatment with −172.3 (mmol/kg DM) DCAD could prevent hypocalcemia compared to a control DCAD of +448.6 (mmol/kg DM). The DCAD level has been an important parameter in the process of formulating diets fed to Holstein cows in recent years [3,4,5,6,7]. Studies have shown that diets with a lower DCAD could enhance health and extend the economic life of transition mammary animals [8,9,10]. Recently, attention has been given to the interaction of DCAD with calcium (Ca) [11], cholecalciferol/calcidiol [12], 5-hydroxy-l-tryptophan [13], vitamin D [14,15,16], and the duration [17,18] of blood Ca level increases. Results of these studies indicated that lower DCAD in association with the above factors was effective in improving homeostasis of peripheral blood Ca. Most recently, Santos et al. [19] and Lean et al. [20] used meta-analyses to determine the effects of varying DCAD on the performance, production, and health of cows; they concluded that a negative DCAD prepartum would increase blood total Ca level at calving and result in fewer disease events.

*Butyrivibrio fibrisolvens*, *Fibrobacter succinogenes*, *Ruminococcus flavefaciens*, and *Ruminococcus albus* were reported as the 4 most important cellulolytic bacteria for digestion and utilization of fiber in the rumen [21,22,23]. However, to our knowledge, there is less information on rumen fermentation, cellulolytic bacteria populations, and microbiota for female goats fed a negative DCAD diet. Based on the strong buffering capability (BC) of ruminal fluid, we hypothesized that a negative DCAD would not affect the rumen fermentation parameters. Therefore, the present study was conducted to evaluate the effect of a negative DCAD diet on rumen pH, BC, volatile fatty acid (VFA) of acetic acid, propionic acid, butyric acid, sum of acetic acid, propionic acid, butyric acid, and acetic acid/propionic acid (A/P), as well as ruminal cellulolytic bacteria populations of *Butyrivibrio fibrisolvens*, *Fibrobacter succinogenes*, *Ruminococcus flavefaciens*, and *Ruminococcus albus*, and microbiota. Growth performance, fluid acid–base balance, and plasma Ca level were also measured. The results should provide a further evaluation on the feasibility of feeding a negative DCAD diet to goats.

## 2. Materials and Methods

### 2.1. Animals and Experimental Design

The animal treatment procedures were approved by the Guizhou University Committee of Experimental Animal Ethics with the number code EAE-GZU-2020-P016. Using a completely randomized block design, 18 Qianbei miscellaneous-color female goats (a native goat breed in the southwest of China) at around the 30th day of pregnancy with a similar body weight (BW; 30.07 kg, SD = 0.55) and age (13 months) were blocked to 3 treatments of 6 replicates with 1 goat per replicate. Animals were fed 1 of 3 diets with varying DCAD levels (mmol/kg DM): +350 (high DCAC; HD), +100 (control; CON), and −150 (low DCAD; LD). The diet was pelleted as total mixture ration (TMR) with a ratio of concentrate to roughage at 30:70. The NaHCO_3_ and NH_4_Cl was included to increase and reduce DCAD for HD and LD, respectively.

Goats were fed in their individual metabolic cages during the whole experiment. The experiment duration was 45 d including a 30-d adaption period and 15-d trial period. The adaption period was divided into 3 stages. In the first stage (1–12 d), goats were observed for health conditions, treated for parasites, and disinfected. During the second stage (13–18 d), goats were allowed to adjust to their respective diets. In the third stage (19–30 d), steady dry matter intake (DMI) was measured for individual goats. After that, in the trial period (31–45 d), goats were strictly fed the treatment diets according to the established DMI at 09:00 and 18:00. All goats had free access to water during the whole experimental process. Ingredients and chemical components of diets are shown in Table 1. The DCAD levels were measured as +338, +152, and −181 for HD, CON, and LD, respectively.

### 2.2. Sample and Measurement

Diet samples were collected daily during 19–45 d and were composited and dried at 65 °C and then ground to pass a 1-mm screen for proximate chemical composition determination of DM, crude protein (CP), crude ash (Ash) [24], neutral detergent fiber (NDF), and acid detergent fiber (ADF) [25]. An atomic absorption spectrophotometer (iCE 3000 SERIES, Thermo Fisher Scientific, USA) was used to measure Na and K contents. Silver nitrate titration was used to determine Cl concentration. The S level was determined using the magnesium nitrate method as previously described [26]. The DCAD was calculated using the following equation according to Block [2]:DCAD = Na (%)/0.0023 + K (%)/0.0039 − Cl (%)/0.00355 − S (%)/0.0016

All goats were weighed on day 32 as the initial body weight and on day 46 as the final body weight. The DMI was recorded daily for each goat calculated by allowance of refusals. The average net body gain (ANG) was determined by subtraction of initial body weight from final body weight. The average daily body gain (ADG) was determined by dividing ANG with the trial period (15 d). The feed conversion ratio (FCR) was the ratio of DMI to ADG.

At 8:30 on day 44, rumen fluid was collected through the esophageal cannula via a vacuum pump (VP30, Labtech Instrument Co. Ltd., Beijing, China) in accordance with the methodology [27]. Rumen pH was measured using a portable type pH meter (PHS-3C, Youke Instrument Co. Ltd., Shanghai, China). Rumen BC was assessed in accordance with Tucker et al. [28]. To determine the rumen VFA, samples were centrifuged at 10,000× *g* for 10 min at 4 °C (Thermo Fisher-ST 16R), and the supernatant fraction filtered through a 0.45-µm filter; at least 1.5 mL supernatant was promptly transferred to a 2 mL centrifuge tube. The 1280 μL of filtrate was mixed with 600 μL of 20% metaphosphoric acid and 120 μL of crotonic acid (internal standard), and the mixtures were centrifuged again under the same conditions. All the steps above are performed on ice. Finally, 1 mL supernatant was transferred to a 2-mL sample vial. The VFA concentrations in filtered samples were determined by gas chromatography (GC-2010-plus, Shimadzu, Japan) equipped with an AOC-20i autosampler, and coupled to a flame ionization detector. The chromatographic separation was performed on a Shimazu SH-Rtx-Wax capillary column (30 m × 0.25 mm × 0.25 μm). Three µL of the sample solution was injected in split mode at a ratio of 50:1. The injection temperature was 200 °C and the detector temperature was 220 °C. The initial temperature of the column was 100 °C for 2 min, increased to 150 °C at a rate of 5 °C per minute. The flow rate was set to 1.08 mL/min. The carrier gas was N_2_ (99.999%) and its pressure was 0.5, H_2_ 0.4, air 0.3~0.4 MPa.

Ruminal fluid samples were stored at −80 °C in aliquots. In order to assure complete breakage of cells for DNA extraction, we performed a bead-beating step before using a Qiagen Mini Stool Kit for DNA extraction. We added 0.25 g sterile 0.1 mm zirconia beads, oscillated for 2 min with the mill, and then performed DNA extraction in accordance with the kit procedure. The DNA was extracted from 200-mg samples using a QIAamp DNA Stool Mini Kit (Qiagen, Hilden, Germany) following the manufacturer’s instructions. The DNA concentration and purity were checked by running samples on 1.0% agarose gels. The PCR amplification of 16S rRNA genes was performed using general bacterial primers: 515F 5′-GTGCCAGCMGCCGCGGTAA-3′ and 926R 5′-CCGTCAATTCMTTTGAGTTT-3′. The primers also contained the Illumina 5′-overhang adapter sequences for the two-step amplicon library building, following the manufacturer’s instructions for the overhang sequences. The initial PCR reactions were carried out in 25-µL reaction volumes with 1–2 µL of DNA template, 250 mM dNTPs, 0.25 mM of each primer, 1×reaction buffer, and 0.5 U of Phusion DNA Polymerase (New England Biolabs, Ipswich, MA, USA). The PCR conditions consisted of initial denaturation at 94 °C for 2 min, followed by 25 cycles of denaturation at 94 °C for 30 s, annealing at 56 °C for 30 s, and extension at 72 °C for 30 s, with a final extension of 72 °C for 5 min. The second step of PCR with dual eight-base barcodes was used for multiplexing. Eight-cycle PCR reactions were used to incorporate two unique barcodes to either end of the 16S amplicons. Cycling conditions consisted of one cycle of 94 °C for 3 min, followed by eight cycles of 94 °C for 30 s, 56 °C for 30 s, and 72 °C for 30 s, and a final extension of 72 °C for 5 min. Prior to library pooling, the barcoded PCR products were purified using a DNA gel extraction kit (Axygen Biotech, Hangzhou, China) and quantified using FTC-3000 TM real-time PCR. The libraries were sequenced by 2 × 300 bp paired-end sequencing on the MiSeq platform using MiSeq v3 Reagent Kit (Illumina) at Tiny Gene Bio-Tech (Shanghai) Co. Ltd.

The raw fastq files were demultiplexed based on the barcode. The PE reads for all samples were run through Trimmomatic (version 0.35) to remove low-quality base pairs using parameters SLIDINGWINDOW: 50:20 and MINLEN: 50. Trimmed reads were then further merged using the FLASH program (version 1.2.11) with default parameters. The low-quality contigs were removed based on screen.seqs command and the singletons were filtered out from the spliced long reads using mothur V.1.39.5 following filtering parameters, maxambig = 0, minlength = 200, maxlength = 580, and maxhomop = 8. The 16S sequences were analyzed using a combination of software mothur (version 1.39.5), UPARSE (usearch version v8.1.1756, http://drive5.com/uparse/, 24 November 2020) and R (version 3.2.3). The demultiplexed reads were clustered at 97% sequence identity into operational taxonomic units (OTUs) using the UPARSE pipeline (http://drive5.com/usearch/manual/uparse cmds.html, 24 November 2020). The OTU representative sequences were used for taxonomic assignment against the Silva 128 database with a confidence score ≥0.6 by the classify.seqs command in mothur. The OTU taxonomies (from phylum to species) were determined based on NCBI. In this study, the mean valid sequence, optimized sequence, and OUT were 38,812, 30,716, and 724, separately. The four members of the rumen cellulolytic bacteria community (*B. fibrisolvens*, *F. succinogenes*, *R. flavefaciens*, and *R. albus*, % of total bacterial 16S rDNA) were selected from “Species” for statistical analysis. OTUs with a similar level of 97% were selected for Venn analysis using R (3.4.1). Rarefaction curves based on observed species and phylogenetic distance whole tree measures plateaued. Rank-abundance distribution curves was performed by ranking OTUs in order of abundance (number of sequences contained) from largest to smallest. Alpha diversity was analyzed using mothur (http://www.mothur.org/wiki/Schloss_SOP#Alpha_diversity, 24 November 2020) and QIIME (http://qiime.org/scripts/alpha_diversity.html?highlight = alpha, 24 November 2020). Principal coordinate analysis (PCoA) was performed, using both weighted and unweighted unique fraction metric (UniFrac) distances that measured the phylogenetic distance between sets of taxa in a phylogenetic tree as the fraction of the branch length of the tree, on the 97% OTU composition and abundance matrix. The microbial community structure was composed of a data table based on species classification information and plotted using R (3.4.1) language GGPLOT2.

Urine pH was measured once every 3 d during the first (1–12 d) and second (13–18 d) adaption periods, once every 2 days during the third stage (19–30 d), and daily for the trial period (31–45 d). Briefly, urine was immediately dipped with special indicator paper (5.4–7.0, SSSreagent Co. Ltd., Shanghai, China; 6.4–9.0, Fuyang Special Paper Co. Ltd., Hangzhou, China) when goats were urinating.

Ten-mL blood samples of every goat were collected from the jugular vein at 45 d and were centrifuged at 805× *g* for 15 min to harvest plasma for analysis of Ca, glucose (Glc), urea nitrogen (UN), alanine aminotransferase (ALT), aspartate transaminase (AST), alkaline phosphatase (AKP), total protein (TP), albumin (Alb), superoxide dismutase (SOD), glutathione peroxidase (GSH-Px), malondialdehyde (MDA), and catalase (CAT).

### 2.3. Statistical Analysis

The MIXED models in SAS 9.4 (SAS Institute Inc, Cary, NC, USA) were applied for analysis of experimental data. A randomized complete block design with repeated measures was used for data analysis. The DCAD levels (+338, +152, and −181) were designated as fixed effects, and goats as the random effect, then Tukey’s method was adopted to determine differences among means of the 3 DCAD treatments. Statistical significance was defined as *p* < 0.05.

## 3. Results

### 3.1. Rumen Fermentation Parameters

There was no significant difference in rumen pH for HD, CON, or LD (*p =* 0.97; Table 2). The variation of DCAD had no effect on ruminal BC, rumen acetic acid, propionic acid, butyric acid, sum of acetic acid, propionic acid, butyric acid, or the A/P levels of the goats (*p* ≥ 0.60).

### 3.2. Rumen Cellulolytic Bacteria

The relative contents of Fibrobacter, Butyrivibrio, Ruminococcus, *F. succinogenes*, *R. flavefaciens*, *B. fibrisolvens*, *R. albus*, and the sum of *F. succinogenes*, *R. flavefaciens*, *B. fibrisolvens,* and *R. albus* were not significantly affected in goats that consumed HD, CON, or LD diets (*p* ≥ 0.12; Table 3).

### 3.3. Sequencing and Diversity of Ruminal Microbiota

After Illumina Miseq high-throughput sequencing, a total of 698,626 valid reads were obtained with an average length of 410 bp. The average valid sequence and optimized sequence of 16 samples were 38,812 and 30,716, separately. The Venn graph showed that there were 1075 identical OTU of total 1261 among 3 groups. There were 22, 25, 5 individual OTUs, accounting for 1.89%, 2.33% and 0.43% for LD, HD, CON, respectively (Figure 1A). Rarefaction curves were established to quantify the OTU coverage of sampling and each rarefaction tended to be gentle with the increase of sequence number (Figure 1B), and meanwhile, the OTU rank abundance in the 3 groups exhibited a gentler slope and wider distribution on the horizontal axis (Figure 1C).

Alpha diversity results showed that DCAD levels did not affect Coverage, Sobs, Chao, Abundance Coverage-based Estimator (Ace), Simpson, Shannon, or PD_whole_tree as listed in Table 4 (*p* ≥ 0.05). Both PCoA weighted UniFrac distance (Figure 1D) and unweighted UniFrac distance (Figure 1E) displayed no obvious microbial community differences between individuals and groups for CON, HD, and LD.

Taxonomic classification summary indicated that 16 phyla were detected in all samples (Figure 2A). At the phylum level, Bacteroidetes (61.60%) was the predominant phylum followed by Firmicutes, Synergistetes, Proteobacteria, Spirochaetae, and Tenericutes with average relative abundances of 25.32%, 5.84%, 1.82%, 2.08%, and 1.2%, respectively. However, there was no difference (*p* ≥ 0.14) among the groups on the above phylum levels except for Firmicutes, which was higher in HD and LD compared to CON (*p* = 0.008, Table 5). 

At the genus level, taxon displayed that the relative abundance of 11 genera were not affected by DCAD among all samples (*p* > 0.05; Figure 2B). At the same time, *Prevotella*, *Paraprevotella*, *Selenomonas*, *Ruminococcus*, *Butyrivibrio*, *Quinella*, *Fretibacterium*, and *Treponema* showed no grouping difference of the genera across treatments (*p* ≥ 0.12). Among the genera that showed more than 0.1% of relative abundance, *Prevotella* was the dominant genus in each group with the highest proportion (Table 5).

### 3.4. Growth Performance

Levels of DMI were unaffected by DCAD variations (*p* = 0.89; Table 6). Lower DCAD had no effect (*p* ≥ 0.63) on growth performance of the final weight, ANG, ADG, or FCR for goats.

### 3.5. Urine pH

There was no difference (*p* = 0.31) in urine pH for HD, CON, or LD during the observation period of 1–12 d, with pH values of 8.48, 8.43, and 8.46, respectively (Figure 3). Urine pH in LD reduced relative to HD and CON (*p* < 0.000) during days 13–18. Urine pH decreased significantly with LD obviously lower (*p* < 0.000) than both HD and CON during days 19–30. During the trial period (31–45 d), LD quadratically reduced urine pH over HD and CON (8.43, 8.42, and 6.75 for HD, CON, and LD, respectively; *p* < 0.000). Urine pH values were unaffected between HD and CON (*p* > 0.05). Furthermore, urine pH had a strong association with DCAD within the trial period (31–45 d; R^2^ = 0.9706; *p* < 0.0001; Figure 4).

### 3.6. Plasma Metabolites

Feeding of the LD diet resulted in the highest plasma Ca level (Table 7), which was higher than both HD and CON (*p* < 0.01). There was no significant difference in other plasma metabolites of Glc, UN, ALT, AST, AKP, TP, Alb, GSH-Px, CAT, SOD, or MDA among the DCAD treatments (*p* ≥ 0.37; Table 7).

## 4. Discussion

In this study, rumen pH was unaffected for 3 DCAD treatments. This was also shown in the study of Apper–Bossard et al. [29]. The reason is that rumen has a strong buffer system to maintain a stable rumen status by keeping any sudden rise or fall in rumen pH within a certain range [30]. Meanwhile, there was no difference for rumen BC for the varying DCAD levels. Church [31] argued that this was because the rumen buffer system was controlled by pH, *p*CO_2_, and VFA. In our study, the lack of difference in rumen BC and pH indicated that negative DCAD would not exert deleterious influence on the rumen internal environment. Thereafter, up to now there has been insufficient information on rumen BC in female goats consuming reduced DCAD, and further study is needed due to its effect on maintaining homeostasis of the rumen. Increasing DCAD, by adding K and Na, had no effect on rumen VFA concentration [32]. Tucker et al. [33] reported that the rumen VFA profile was unaffected by DCAD levels of −100, 0, 100, and 200 mmol/kg DM. Apper–Bossard et al. [29] found no significant difference in VFA concentration with varying DCAD. These results are consistent with our study in which rumen VFA profiles were not significantly affected by DCAD variation. Correspondingly, the rumen A/P level was unaffected by DCAD treatment. This indicates that the rumen fermentation pattern was unchanged by the 3 DCAD treatments. With dairy goats as the experimental animals fed 2 DCAD levels at 349 and −167 mmol/kg DM in our most recent study, it was found that rumen VFAs were also not influenced (unpublished data). Results from the current study show that DCAD variation had no influence on ruminal abundance of *B. fibrisolvens*, *F. succinogenes*, *R. flavefaciens*, or *R. albus*, indicating that negative DCAD would not affect the growth or colonization of rumen cellulolytic bacteria. This may be likely associated with stable rumen pH maintained by the constant ratio of concentrate to roughage (30:70) for the 3 DCAD diets used in this study. In addition, the overall low abundance of cellulolytic bacteria is likely due to the systematic underestimation of the particulate-associated community due to the use of stomach tubing as a sampling method, which oversamples the planktonic community.

Rumen bacterial species, the Chao index, and the Ace index were closely related with rumen pH levels because they could alter the bacterial community structure, which is supported by Guo et al. [34], who found that decreased rumen pH upregulated the bacterial diversity, composition, and abundance of bacteria. As mentioned above, rumen pH showed no treatment effects by DCAD level. This exactly explained why the Chao index, Ace index, Simpson index, and Beta diversity of the 3 groups were homogeneous. The Shannon index of the HD group was the highest, possibly because NaHCO_3_ was added as a buffer to neutralize gastric acid and was essential for stomach health by creating a suitable internal environment for rumen microorganisms. In addition, the dominant rumen bacteria of the 3 groups were Bacteroidetes followed by Firmicutes and Synergistetes, which coincided with the results of a previous study in which the abundance of rumen bacteria is ordered by Bacteroidetes, Firmicutes, and Synergistetes at the level of phylum [35]. In terms of genus level, the relative abundance of *Prevotella*, one of the primary carbohydrate-degrading microorganisms, was the highest, which was supported by the previous study [36]. 

It is well known that rumen pH, BC, VFA, cellulolytic bacteria population, and rumen microbiota diversity are crucial in the fermentation status of goats. The unaffected parameters in the present study provide the feasibility of feeding a negative DCAD diet to female goats from the aspects of rumen fermentation.

The level of feed intake is the most important prerequisite for animal growth performance. Generally, pure anionic salt would reduce feed intake when it was simply mixed into the diet, due to its bitter taste and poor palatability [37]. Therefore, improving palatability of anionic salt is important for DMI and growth performance. Our previous study [38] showed that DMI of female goats fed a negative DCAD did not decrease because the anionic salts were mixed with molasses and dried distillers grains with solubles. Takagi and Block [39,40,41] also observed that reducing DCAD did not impact DMI containing anionic salts when feeding a total mixture ration. Diets were pelleted in the study, and therefore, they were unaffected for goats fed diet HD, CON, and LD. This indicates that DMI is unaffected by anionic salts’ inclusion as long as the bitter taste is concealed. The levels of DCAD had no effect on final body weights. This can be attributed to the similar DMI level and possibly because the female goats were a local breed, with their adult steady body weight averaging as much as 35 kg, and thus there was limited potential for body weight gain. Accordingly, ADG and FCR did not show differences among HD, CON, or LD treatments.

Urine pH is a useful indicator to monitor the effect of a reduced DCAD diet on acid–base balance in goats, sheep [42], and dairy cows [38,43]. This phenomenon can be explained using the strong ion difference theory [44], which argued that with reductions of DCAD, the concentration of anions in blood would increase and cause the kidney to expel redundant H^+^ in urine, resulting in lower urine pH. The recommended urine pH is 6.5–6.8, because a urine pH level too low would exert a burden on the kidneys [45,46,47]. Our results showed that urine pH value in goats fed LD was lower than HD and CON. This is accordant with the recommended level, and there is a strong association between DCAD and the urine pH in the trial period, suggesting that the LD level is appropriate for the diet of goats.

Muscle contraction, conduction of nervous impulses, and signal transduction are closely dependent on blood Ca homeostasis. Following the study of Block [2] and subsequent results of ruminant researchers [48,49], reducing the DCAD level has been the most commonly used strategy to increase blood Ca levels in transition mammary animals [50]. Our previous study showed that reducing DCAD could increase the plasma Ca concentration of female goats [38]. In the current experiment, the LD caused a higher plasma Ca level than HD and CON by 25.97% and 22.27%, respectively, indicating more stable blood Ca homeostasis. Horst et al. [51] and Goff and Horst [52] claimed this may be because LD-induced acidic status enhanced Ca absorption in the gastro-intestine and also increased Ca resorption in the bone, facilitating Ca matrix flow into blood for easier transfer from lumen to blood.

Blood measurements are useful to reflect the metabolic status of animals. Our results showed that all plasma levels of Glc, UN, ALT, AST, AKP, TP, Alb, SOD, GSH-Px, MDA, and CAT were unaffected by DCAD variation. This is supported by a previous study of Melendez and Poock [9], who reported that lowering DCAD had little effect on blood Alb. Our previous study also found that DCAD levels (+300, +150, 0, and −150) had no significant effect on plasma GSH-Px or MDA content in female goats [38].

## 5. Conclusions

In conclusion, negative DCAD has no deleterious influence on rumen fermentation parameters and rumen microbiota, showing no harm to rumen fermentation of female goats. The blood Ca level is increased and urine pH is decreased by DCAD reduction. These results may provide the feasibility of feeding a negative DCAD to female goats.

## Figures and Tables

**Figure 1 animals-11-00664-f001:**
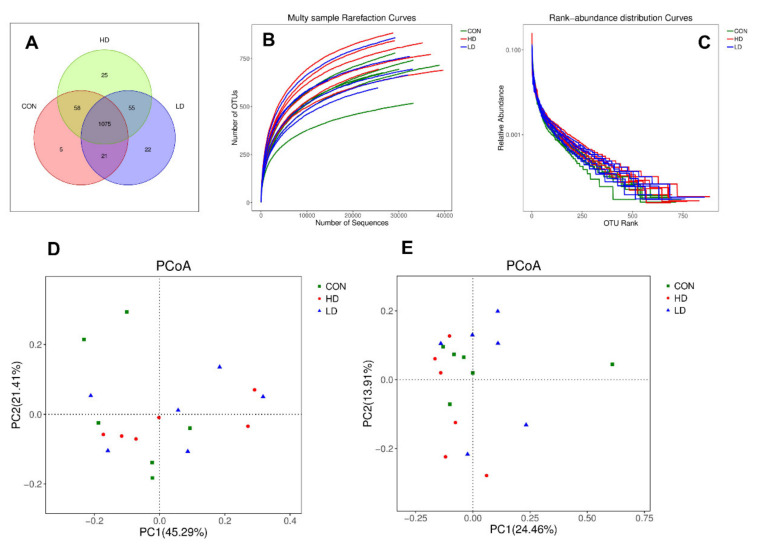
Rumen bacterial diversity of female goats fed varying dietary cation-anion difference. (**A**) The Venn graph. (**B**) Rarefaction curves. (**C**) Rank abundance distribution curves. (**D**) Principal coordinate analysis (PCoA) based on weighted UniFrac distance. (**E**) PCoA based on unweighted UniFrac distance.

**Figure 2 animals-11-00664-f002:**
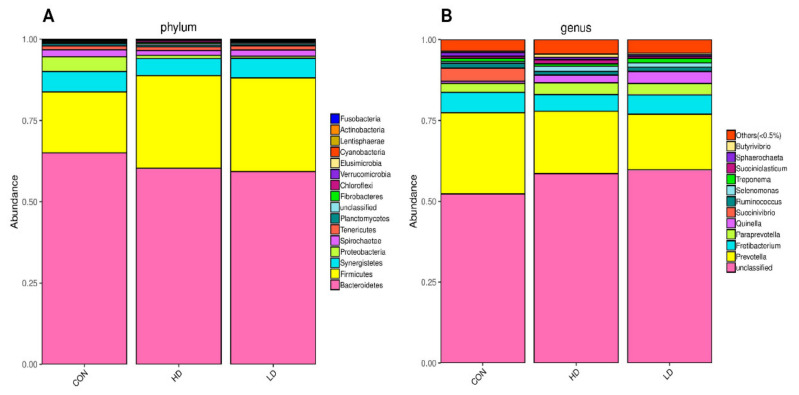
Rumen bacterial diversity and taxonomic classification of female goats fed varying dietary cation-anion difference. (**A**) Distributions of rumen microbiota at phylum level. (**B**) Distributions of rumen microbiota at genus level.

**Figure 3 animals-11-00664-f003:**
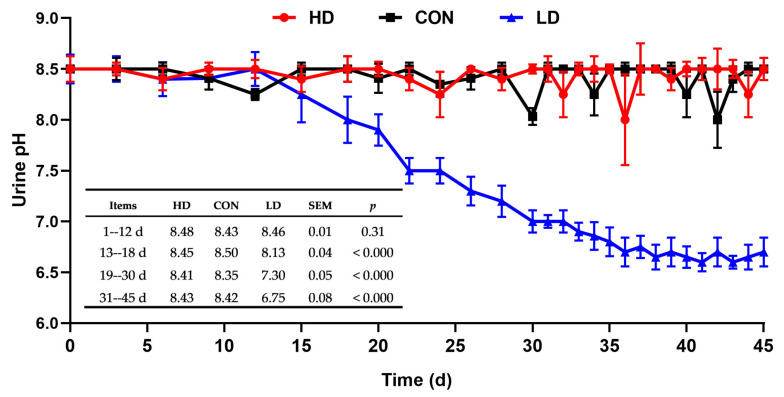
Urine pH variation of female goats fed varying dietary cation-anion difference throughout the experiment.

**Figure 4 animals-11-00664-f004:**
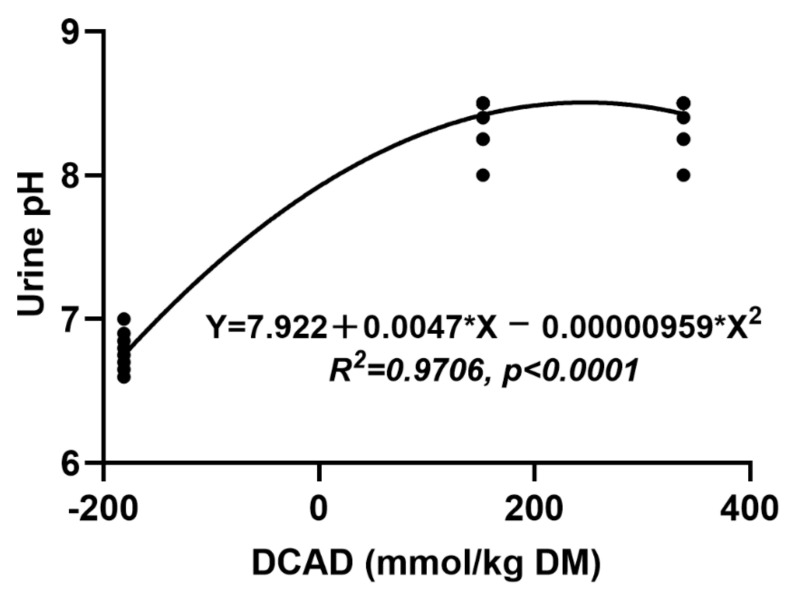
Association between urine pH and varying dietary cation-anion difference of female goats within the trial period.

**Table 1 animals-11-00664-t001:** Ingredients and chemical components of diet for goat (%, dry matter (DM) basis).

Items	DCAD ^1^		
	HD	CON	LD
Ingredients			
Concentrate ^2^	30	30	30
Peanut straw	50	50	50
Faba bean straw	20	20	20
NaHCO_3_ (g/d)	12	--	--
NH_4_Cl (g/d)	--	--	15
Chemical components ^3^			
Dry matter	93.70	93.77	93.90
Crude protein	13.40	13.64	14.21
Neutral detergent fiber	39.40	39.64	39.82
Acid detergent fiber	30.35	30.62	30.64
Crude ash	11.98	12.97	12.29
Na	0.52	0.19	0.14
K	1.08	0.98	0.87
Cl	0.23	0.27	1.25
S	0.16	0.17	0.18
DCAD (mmol/kg DM)	+338	+152	−181

^1^ DCAD: Dietary cation-anion difference; HD: High DCAD; CON: Control; LD: Low DCAD. ^2^ Composition and proportion (%): Corn 51.07, soybean meal 24.82, wheat bran 11.19, rapeseed meal 8.07, CaHPO_4_ 0.45, lysine 0.27, methionine 0.29, NaCl 0.77, and premix 3.06. ^3^ Actually measured values.

**Table 2 animals-11-00664-t002:** Effect of varying dietary cation-anion difference on the rumen fermentation parameters of female goats.

Items	DCAD ^1^			SEM ^2^	*p*-Value
	HD (+338)	CON (+152)	LD (−181)		
pH	7.2	7.2	7.2	0.14	0.97
BC ^3^ (mL/L)	44.7	43.3	43.1	1.38	0.67
Acetic acid (mmol/L)	42.2	43.4	44.8	2.32	0.74
Propionic acid (mmol/L)	13.0	13.1	12.7	0.84	0.95
Butyric acid (mmol/L)	8.0	8.5	8.1	0.83	0.91
Sum of Ac + Pro + But ^4^ (mmol/L)	63.2	65.0	65.6	3.41	0.87
A/P ^5^	3.25	3.31	3.53	0.29	0.60

^1^ DCAD: Dietary cation-anion difference; HD: High DCAD; CON: Control; LD: Low DCAD. ^2^ SEM: Standard error of mean. ^3^ BC: Buffering capability. ^4^ Ac = Acetic acid; Pro: Propionic acid; But: Butyric acid. ^5^ A/P: Acetic acid/propionic acid.

**Table 3 animals-11-00664-t003:** Effect of varying dietary cation-anion difference on the relative abundance (%) of rumen cellulolytic bacteria communities of female goats.

Genus	Species	DCAD ^1^			SEM ^2^	*p*-Value
		HD (+338)	CON (+152)	LD (−181)		
*Fibrobacter* (%)		0.444	0.362	0.326	0.10	0.69
	*F. succinogene* (%)	0.443	0.362	0.324	0.10	0.70
*Butyrivibrio* (%)		1.015	0.459	0.478	0.13	0.12
	*B. fibrisolvens* (%)	0.12	0.085	0.065	0.03	0.59
*Ruminococcus* (%)		1.054	1.538	1.276	0.27	0.79
	*R. flavefaciens* (%)	0.343	0.414	0.119	0.21	0.51
	*R. albus* (%)	0.005	0.018	0.007	0.01	0.50
	Sum of *F. succinogenes*, *R. flavefaciens*, *B. fibrisolvens and R. albus*	0.911	0.879	0.515	0.16	0.57

^1^ DCAD: Dietary cation-anion difference; HD: High DCAD; CON: Control; LD: Low DCAD. ^2^ SEM: Standard error of mean.

**Table 4 animals-11-00664-t004:** Effect of varying dietary cation-anion difference on rumen bacterial community richness and diversity of female goats.

Items	DCAD ^1^			SEM ^2^	*p*-Value
	HD (+338)	CON (+152)	LD (−181)		
Coverage (%)	99.52	99.49	99.45	0.06	0.96
Sobs	785	686	701	35.90	0.29
Chao	906	843	811	4.28	0.33
Ace ^3^	900	834	815	4.33	0.36
Simpson	0.027	0.033	0.028	0.04	0.76
Shannon	4.87	4.45	4.69	0.21	0.05
PD_whole_tree	55.05	49.90	50.42	1.08	0.10

^1^ DCAD: Dietary cation-anion difference; HD: High DCAD; CON: Control; LD: Low DCAD. ^2^ SEM: Standard error of mean. ^3^ Ace: Abundance Coverage-based Estimator.

**Table 5 animals-11-00664-t005:** Effect of varying dietary cation-anion difference on relative abundance (%) of bacteria taxa > 0.1% of average abundance in the rumen fluid of female goats.

Phylum	Genus	DCAD ^1^			SEM ^2^	*p*-Value
		HD (+338)	CON (+152)	LD (−181)		
Bacteroidetes		60.4	65.1	59.3	2.81	0.33
	*Prevotella*	18.9	24.2	16.1	3.05	0.22
	*Paraprevotella*	3.58	2.8	3.5	1.24	0.88
Firmicutes		28.4 ^a^	18.7 ^b^	28.8 ^a^	2.23	0.008
	*Selenomonas*	1.6	0.6	1.3	0.78	0.63
	*Ruminococcus*	1.1	1.5	1.3	0.49	0.79
	*Succiniclasticum*	1.3	0.7	0.7	0.25	0.14
	*Butyrivibrio*	1.0	0.5	0.5	0.20	0.12
	*Quinella*	2.3	0.7	3.7	1.27	0.28
Synergistetes		5.2	6.3	6.0	2.03	0.92
	*Fretibacterium*	5.2	6.3	6.0	2.04	0.92
Spirochaetae		1.4	2.1	1.9	0.49	0.61
	*Treponema*	0.7	1.0	1.4	0.37	0.46
Proteobacteria		1.1	4.5	0.7	1.40	0.14
Tenericutes		1.1	1.2	1.3	0.29	0.82

^a,b^ Means in the superscript differs in the same row (p < 0.05). ^1^ DCAD: Dietary cation-anion difference; HD: High DCAD; CON: Control; LD: Low DCAD. ^2^ SEM: Standard error of mean.

**Table 6 animals-11-00664-t006:** Effect of varying dietary cation-anion difference on growth performance of female goats.

Items	DCAD ^1^			SEM ^2^	*p*-Value
	HD (+338)	CON (+152)	LD (−181)		
Initial body weight (kg)	30.3	30.4	29.4	1.36	0.86
Final body weight (kg)	31.3	31.5	30.4	1.48	0.82
DMI ^3^ (g/d)	899.0	857.5	864.0	64.85	0.89
ANG (kg)	1.0	1.1	1.0	0.21	0.89
ADG (g/d)	65.3	71.3	64.7	8.28	0.63
FCR	13.8	12.0	13.4	3.00	0.81

^1^ DCAD: Dietary cation-anion difference; HD: High DCAD; CON: Control; LD: Low DCAD. ^2^ SEM: Standard error of mean. ^3^ DMI: Dry matter intake; ANG: Average net body gain; ADG: Average daily body gain; FCR: Feed conversion ratio (the ratio of DMI to ADG).

**Table 7 animals-11-00664-t007:** Effect of varying dietary cation-anion difference on the plasma metabolites of female goats.

Items	DCAD ^1^			SEM ^2^	*p*-Value
	HD (+338)	CON (+152)	LD (−181)		
Ca (mmol/L)	2.3 ^b^	2.4 ^b^	2.9 ^a^	0.08	<0.01
Glc ^3^ (mmol/L)	4.6	5.1	4.6	0.34	0.56
UN (mmol/L)	6.2	5.6	6.2	0.41	0.50
ALT (IU/L)	8.6	10.2	10.8	1.70	0.85
AST (IU/L)	11.2	14.0	11.9	2.01	0.80
AKP(King unit/100 mL)	20.8	21.7	17.8	3.75	0.75
TP (g/L)	99.4	112.9	92.6	11.47	0.37
Alb (g/L)	36.4	37.1	35.2	2.35	0.85
GSH-Px (U/mL)	670.6	755.3	709.4	152.50	0.59
CAT (U/mL)	2.3	1.9	2.5	0.66	0.84
SOD (U/mL)	68.5	63.6	68.0	3.38	0.43
MDA (nmol/mL)	37.1	34.6	36.8	2.54	0.75

^a,b^ Means in the superscript differs in the same row (*p* < 0.05). ^1^ DCAD: Dietary cation-anion difference; HD: High DCAD; CON: Control; LD: Low DCAD. ^2^ SEM: Standard error of mean. ^3^ Glc: Glucose; UN: Urea nitrogen; ALT: Alanine aminotransferase; AST: Aspartate transaminase; AKP: Alkaline phosphatase; TP: Total protein; Alb: Albumin; SOD: Superoxide dismutase; GSH-Px: Glutathione peroxidase; MDA: Malondialdehyde; CAT: Catalase.

## Data Availability

Data sharing not applicable.

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
