# Peer review of "Feeding a Negative Dietary Cation-Anion Difference to Female Goats Is Feasible, as Indicated by the Non-Deleterious Effect on Rumen Fermentation and Rumen Microbial Population and Increased Plasma Calcium Level"

_animals, 2021, doi:10.3390/ani11030664_

Round 1
Reviewer 1 Report
The authors have made many changes to improve their manuscript. The reviewer notes that the authors addressed all of the Specific comments raised in the review of the original manuscript, but did not substantively address the two major criticisms of the work (in the “Broad comments” in the original review). The reviewer repeats those comments here:
“The reviewer has major concerns with the microbiota methodologies used by the authors. First, the rumen samples were collected using a stomach tube, which would yield mostly rumen liquid proportionally deficient in feed particles on which most bacteria --particularly the cellulolytic bacteria -- reside. Second, the authors used a Qiagen Stool kit for DNA recovery. Kit based methods (and this kit in particular) have been shown to provide low recovery of microbial DNA (see Table 2 of Henderson et al. 2013, doi: 10.1371/journal.pone.0074787 ), and whose composition does not represent well the bacterial community (see Table 3 of that report). These biases contributed to an unusually low overall abundance of the four major cellulolytic bacterial species (Table3), which in aggregate never exceeded 1% of the bacterial community. The DNA isolation method used thus had the net effect of weakening the study, and the authors need to explicitly justify/discuss this in the manuscript. Failure to do so has the net effect of perpetuating such poor choices by other researchers.”
At the end of their response to reviewer, the authors addressed the first criticism as follows:
“Author response: We think it would have little effect in the study. The first reason is that this ruminal fluid sampling method has been applied in previous study (Wang et al., 2016). The second reason is that we had consecutively fed goats for 14 (31st-44th) days before sampling ruminal samples and the rumen internal environment has been steady. The third reason is that the stomach tube was inserted into rumen center to make the ruminal fluid and feed particles blending for obtain representative rumen fluid samples. Wang, M.; Wang, R.; Janssen, P.H.; Zhang, X.M.; Sun, X.Z.; Pacheco, D.; Tan, Z.L. Sampling procedure for the measurement of dissolved hydr “ [the text end here].
The reviewer finds this inadequate. First, just because the authors got away with publishing an inferior method earlier does not make the method any better. Second, the argument involving a steady rumen environment has no bearing on the adequacy of the sampling method. Third, placement of the stomach tube is not relevant, as it does not change the fact that the limitation with stomach tubing is that it preferentially selects against particulate sampling. The reviewer realizes that lack of access via an cannula is common in ruminant studies, and should not serve as a sole justification for invalidating a study. Nevertheless, the authors need to make the reader aware that it does represent a limitation that can affect certain results, viz., the recovery of particulate-bound bacteria, especially the cellulolytics. So, the reviewer suggests that the authors include in the manuscript an explicit statement (when discussing the results in Table 3), that the overall low abundance of cellulolytic bacteria is likely due to systematic underestimation of the particulate-associated community due to the use of stomach tubing as a sampling method, which oversamples the planktonic community.
Regarding the second criticism, on the DNA extraction method, the authors have aided their case by including text (L146-147) that indicates their use of a bead-beating step prior to use of the Qiagen Mini-Stool kit. This new information (not in the original manuscript) blunts much of the reviewer’s previous criticism. The reviewer suggests modifying the text to state, “In order to assure complete breakage of cells for DNA extraction, we performed a bead-beating step before using a Qiagen Mini Stool kit for DNA extraction …….”
Additional comments:
L4:Change “is Feasible to Female Goats” to “to Female Goats is Feasible, as”.
L5: Change “Undeletrious” to “Non-deleterious”.
L73-74, L81-82, L263, L320-321, L336: Italicize genus and species names.
L77: The strong buffering capacity of what?
Table 2: A/P can expressed to an additional significant figure (0.01).
L226: Insert “relative abundance (%) of” ahead of “rumen”.
L294 and Table 7: Change abbreviation for glucose from Glu to Glc (to avoid confusion with the common abbreviation for glutamate).
L328: Change “was lack of” to “showed no”.
L334: Change “was coincide” to “coincided”.
Author Response
Q1. The reviewer finds this inadequate. First, just because the authors got away with publishing an inferior method earlier does not make the method any better. Second, the argument involving a steady rumen environment has no bearing on the adequacy of the sampling method. Third, placement of the stomach tube is not relevant, as it does not change the fact that the limitation with stomach tubing is that it preferentially selects against particulate sampling. The reviewer realizes that lack of access via an cannula is common in ruminant studies, and should not serve as a sole justification for invalidating a study. Nevertheless, the authors need to make the reader aware that it does represent a limitation that can affect certain results, viz., the recovery of particulate-bound bacteria, especially the cellulolytics. So, the reviewer suggests that the authors include in the manuscript an explicit statement (when discussing the results in Table 3), that the overall low abundance of cellulolytic bacteria is likely due to systematic underestimation of the particulate-associated community due to the use of stomach tubing as a sampling method, which oversamples the planktonic community.
Author response: Thanks for your suggestion. We would include explicit statements in the discussion of the revised manuscript, as follows: In addition, the overall low abundance of cellulolytic bacteria is likely due to the systematic underestimation of the particulate-associated community due to the use of stomach tubing as a sampling method, which oversamples the planktonic community. (see lines 374-376 in blue color)
Q2. Regarding the second criticism, on the DNA extraction method, the authors have aided their case by including text (L146-147) that indicates their use of a bead-beating step prior to use of the Qiagen Mini-Stool kit. This new information (not in the original manuscript) blunts much of the reviewer’s previous criticism. The reviewer suggests modifying the text to state, “In order to assure complete breakage of cells for DNA extraction, we performed a bead-beating step before using a Qiagen Mini Stool kit for DNA extraction …….”
Author response: Thanks for your suggestion. We had modified the text to state, “In order to assure complete breakage of cells for DNA extraction, we performed a bead-beating step before using a Qiagen Mini Stool kit for DNA extraction.” (see lines 148-150 in blue color)
Q3. L4:Change “is Feasible to Female Goats” to “to Female Goats is Feasible, as”.
Author response: Thanks for your question. We have changed it in the revised manuscript. (see line 5 in blue color)
Q4. L5: Change “Undeletrious” to “Non-deleterious”.
Author response: Thanks for your question. We have changed it in the revised manuscript. (see lines 5-6 in blue color)
Q5. L73-74, L81-82, L263, L320-321, L336: Italicize genus and species names.
Author response: Thanks for your question. We Italicized all genus and species in the revised manuscript. (see lines 37-38, 69, 77-78, 223-224, 265-268, 387 in blue color)
Q6. L77: The strong buffering capacity of what?
Author response: Thanks for your question. It refers to the buffering capacity of ruminal fluid. We would change “The strong buffering capacity” to “The strong buffering capacity (BC) of ruminal fluid” in the revised manuscript. (see line 73 in blue color)
Q7. Table 2: A/P can expressed to an additional significant figure (0.01).
Author response: Thanks for your question. There is no difference for A/P in the manuscript, so we think it is unnecessary to express the additional significant figure, and meanwhile, we changed “3.3, 3.3 and 3.7” to “3.25, 3.31 and 3.53” in the revised manuscript. (see line 216 in blue color)
Q8. L226: Insert “relative abundance (%) of” ahead of “rumen”.
Author response: Thanks for your question. We would add it in the revised manuscript. (see line 230 in blue color)
Q9. L294 and Table 7: Change abbreviation for glucose from Glu to Glc (to avoid confusion with the common abbreviation for glutamate).
Author response: Thanks for your question. We have changed all of Glu to Glc in the revised manuscript. (see lines 201, 334, 346 in blue color)
Q10. L328: Change “was lack of” to “showed no”.
Author response: Thanks for your question. We have changed “was lack of” to “showed no” in the revised manuscript. (see line 380 in blue color)
Q11. L334: Change “was coincide” to “coincided”.
Author response: Thanks for your question. We have changed “was coincide” to “coincided” in the revised manuscript. (see line 385 in blue color)
Please see the attachment.

Reviewer 2 Report
The document has improved, but there are some issues that should be solved.
Line 60: Delete “of” in “… (Cl- and S2-) of per kg of DM…”.
Material and methods:
Now it is clear that goats were fed in individual metabolic cages. Were the animals allocated in the metabolic cages during the whole experiment? Authors should clarify this point and state it clearly in the document.
Table 1: Authors should add the superscript 3 in the first row. HD, CON and LD should be defined. For example: “Ingredients and chemical components of the diets (high dietary cation anion difference, HD, control, CON and low dietary cation anion difference, LD. for goat (%, DM basis)”.
Statistical analysis: To analyze the effect of the DCAD levels on the different OTUs the Kruskal-Wallis test is recommended. If possible, authors should perform this test, or at least authors should state that they had performed a test to check the normality of the data.
Results:
Tables should be self-explanatory. Every abbreviation in each table should be defined. In tables 2 to 7 SEM should be defined and have one decimal more than the mean value.
Line 234: OTU instead of OUT
Line 240 and figure 1: As suggested in the previous review, it is clearer to include the word “distance” after UniFrac.
Authors should check the genera names. They should be in italics. (see lines 260-263)
Discussion:
The last paragraph added by the authors is not what I meant. It would be interesting an introductory paragraph remembering the aim of the study and how was performed.
Author Response
Q1. Line 60: Delete “of” in “… (Cl- and S2-) of per kg of DM…”.
Author response: Thanks for your question. We have deleted “of” in “… (Cl- and S2-) of per kg DM…”in the revised manuscript. (see line 56 in blue color)
Q2. Now it is clear that goats were fed in individual metabolic cages. Were the animals allocated in the metabolic cages during the whole experiment? Authors should clarify this point and state it clearly in the document.
Author response: Thanks for your question. The animals were allocated in the metabolic cages during the whole experiment. We would change it in the revised manuscript. (see line 92 in blue color)
Q3. Table 1: Authors should add the superscript 3 in the first row. HD, CON and LD should be defined. For example: “Ingredients and chemical components of the diets (high dietary cation anion difference, HD, control, CON and low dietary cation anion difference, LD. for goat (%, DM basis)”.
Author response: Thanks for your suggestion. We added the superscript 1 in the first row, and HD, CON and LD have been defined in Table 1. (see line 111 in blue color)
Q4. Statistical analysis: To analyze the effect of the DCAD levels on the different OTUs the Kruskal-Wallis test is recommended. If possible, authors should perform this test, or at least authors should state that they had performed a test to check the normality of the data.
Author response: The Kolmogorov-Smirnov test was adopted to check the normality of the data, whose result showed p = 0.20 > 0.05, so the data is normally distributed. In addition, the homogeneity test of variance showed p = 0.930 > 0.05. Therefore, the Kruskal-Wallis test, a non-parametric statistical method, was not used.
Q5. Tables should be self-explanatory. Every abbreviation in each table should be defined. In tables 2 to 7 SEM should be defined and have one decimal more than the mean value.
Author response: Thanks for your suggestion. Every abbreviation in each table have been defined. (see Table 2-7 in blue color)
Q6. Line 234: OTU instead of OUT
Author response: Many thanks. We have changed it in the revised manuscript. (see line 239 in blue color)
Q7. Line 240 and figure 1: As suggested in the previous review, it is clearer to include the word “distance” after UniFrac.
Author response: Many thanks. We would add it in the revised manuscript. (see lines 41, 245, 251 in blue color)
Q8. Authors should check the genera names. They should be in italics. (see lines 260-263)
Author response: Thanks for your question. We Italicized all genus and species in the revised manuscript. (see lines 37-38, 69, 77-78, 223-224, 265-268, 387 in blue color)
Q9. The last paragraph added by the authors is not what I meant. It would be interesting an introductory paragraph remembering the aim of the study and how was performed.
Author response: Thanks for your suggestion. It seems that we had misunderstood your words. We think there is no urgent need to add the introductory paragraph remembering the aim of the study and how was performed in the last paragraph, as it has been displayed in the abstract and material and method section. Also, we would delete the last paragraph in the revised manuscript.

Reviewer 3 Report
Review of Manuscript Animals-1111935
The aim of the manuscript was to evaluate the effect of dietary DCAD levels on ruminal fermentation parameters, ruminal microbial population and Ca plasma level. This was a much elaborated experiment that shows that DCAD do not affect ruminal fermentation parameters or microbiota in the rumen. However, these results are not really surprising. I would like to request the authors to consider the following comments and remarks:
Major comments
The introduction explains well the background and importance of DCAD for dairy cows and properly justified the realization of the present experiment.
M&M is described with sufficient detail and results are mostly clearly presented
Minor comments
L215: Instead of simply stating p>0.05, please give always the obtained p-values. By this, each reader can make its own interpretations. In this case p =0.97. Please consider this comment and check along the paper
L218: p≥0.60
L266: Enlarge the figure to improve the visibility/quality of the results
L277: p=0.31
L279: Decreased compared to what? This is not clear. “Dramatically” is exaggerated. This p-value refers to the differences in treatments during d 13-18 and not to a decrease from d 1-12 to d 13-18. A decrease of around 0.32 to 0.37 in pH is not too dramatic. Rewrite the sentence
L279: Preferable do not use the word “decreased”. Can be confounded when looking the figure. Instead make the remark e.g. that with treatment ”x “ was lower than with treatment “y “
L283: quadratic
L305: Block of goats? Confusing
L302-305: Or maybe just because the total amount of minerals (Na, K, Cl, S) is relatively low compared to the rumen volume and content to affect pH?
Author Response
Q1. L215: Instead of simply stating p>0.05, please give always the obtained p-values. By this, each reader can make its own interpretations. In this case p =0.97. Please consider this comment and check along the paper
Author response: Thanks for your suggestion. We have changed some of the P values in the revised manuscript. (see lines 213, 225, 244, 261, 267, 295-296, 309 in blue color)
Q2. L218: p≥0.60
Author response: Many thanks. We have changed it in the revised manuscript. (see line 215 in blue color)
Q3. L266: Enlarge the figure to improve the visibility/quality of the results
Author response: Many thanks. We have enlarged the figure 2 in the revised manuscript. (see line 269 in blue color)
Q4. L277: p=0.31
Author response: Many thanks. We have changed it in the revised manuscript. (see line 309 in blue color)
Q5. L279: Decreased compared to what? This is not clear. “Dramatically” is exaggerated. This p-value refers to the differences in treatments during d 13-18 and not to a decrease from d 1-12 to d 13-18. A decrease of around 0.32 to 0.37 in pH is not too dramatic. Rewrite the sentence
Author response: Thanks for your question. We changed “Urine pH decreased dramatically (p < 0.000) during 13–18 d” to “Urine pH in LD reduced relative to HD and CON (p < 0.000) during 13–18 d” in the revised manuscript. (see lines 310-311 in blue color)
Q6. L279: Preferable do not use the word “decreased”. Can be confounded when looking the figure. Instead make the remark e.g. that with treatment ”x “ was lower than with treatment “y “
Author response: Thanks for your question. We would change “Urine pH decreased dramatically (p < 0.000) during 13–18 d” to “Urine pH in LD reduced relative to HD and CON (p < 0.000) during 13–18 d” in the revised manuscript. (see lines 310-311 in blue color)
Q7. L283: quadratic
Author response: Thanks for your question. We would change “During the trial period (31–45 d), compared with HD and CON, LD reduced urine pH over HD and CON (8.43, 8.42 and 6.75 for HD, CON and LD, respectively; p < 0.000).” to “During the trial period (31–45 d), LD quadratically reduced urine pH over HD and CON (8.43, 8.42 and 6.75 for HD, CON and LD, respectively; p < 0.000).” in the revised manuscript. (see line 313 in blue color)
Q8. L305: Block of goats? Confusing
Author response: Thanks for your question. We changed “the 3 blocks of goats” to “the varying DCAD levels”. (see line 357 in blue color)
Q9. L302-305: Or maybe just because the total amount of minerals (Na, K, Cl, S) is relatively low compared to the rumen volume and content to affect pH?
Author response: We think it is not the reason, because goats had free access to diet in the whole experiment. We would suggest that the result is related with the rumen buffering capability. (see lines 355-356 in blue color)

Round 2
Reviewer 2 Report
Thank you for answering all the question. And thank you for accepting the suggestions.
So, I consider the manuscript suitable for publication.
This manuscript is a resubmission of an earlier submission. The following is a list of the peer review reports and author responses from that submission.
Round 1
Reviewer 1 Report
REVIEW OF animals-1060283
The authors fed female goats a high-forage diet amended with inorganic salts to achieve three different levels of anion/cation balance, as quantified by DCAD calculations. The different diets resulted in very few changes among a host of production, metabolic and microbial variables, from which the authors concluded the feeding low-DCAD diets – as is commonly done post-partum to avoid hypocalcemia – would have no deleterious effects on the animals.
Overall, although the work is not particularly novel or elegant, the data may of interest to producers. In general, the experiments appear to have been adequately performed, save for a poor choice for the method of community DNA analysis (see Specific comments below). Some improvements in English are warranted (see “Minor edits” below).
Broad comments:
The reviewer has major concerns with the microbiota methodologies used by the authors. First, the rumen samples were collected using a stomach tube, which would yield mostly rumen liquid proportionally deficient in feed particles on which most bacteria --particularly the cellulolytic bacteria -- reside. Second, the authors used a Qiagen Stool kit for DNA recovery. Kit based methods (and this kit in particular) have been shown to provide low recovery of microbial DNA (see Table 2 of Henderson et al. 2013, doi: 10.1371/journal.pone.0074787 ), and whose composition does not represent well the bacterial community (see Table 3 of that report). These biases contributed to an unusually low overall abundance of the four major cellulolytic bacterial species (Table 3), which in aggregate never exceeded 1% of the bacterial community. The DNA isolation method used thus had the net effect of weakening the study, and the authors need to explicitly justify/discuss this in the manuscript. Failure to do so has the net effect of perpetuating such poor choices by other researchers.
Preparing a review report is made difficult by the lack of line numbers in the manuscript.
Specific comments:
P3, paragraph 1, L1: Please provide name of the organizational animal control committee, and the protocol number.
P3, L2-4: What is meant by “miscellaneous”? The reviewer is (and most readers likely are) unfamiliar with this breed, which appears to be of small frame, so it would be useful to add a sentence here further describing the breed. Is it primarily used for dairy, or meat, or both? What was the pregnancy status of the does during the course of the experiment? It would help if the authors would indicate what motivated the selection of these particular animals and growth stage for the experiment.
P4, para.1, L3-4: Were amylase and sodium sulfite added to the neutral detergent solution?
P4, para.3, L8-11: Provide more detail on the chromatography (carrier gas flow rate, column dimensions, column temperature program).
P5, para. 1:It seems odd that the authors would use a relatively qualitative method of pH measurement (pH indicator paper) when they had available to them a pH meter.
P5. para.4: Were singletons (which often arise from sequencing errors) removed from the analysis? Beyond the four species-level analyses, were any genus-level analyses conduced for Fibrobacter or Ruminococcus?
P5, last line: It appears that “Total VFA” was actually the sum of only acetate, propionate and butyrate, and not of other C4 acids or C5 acids. As these would have been chromatographically separated, why were their concentrations not included.
P6, Section 3.3: What was the average number of sequences obtained across samples? The number of OTUs is quite small compared to those in other ruminal studies using the Illumina sequencing platform. Again, this may reflect low DNA yields.
P7, para.1: The rarefaction curves are useful, but the authors should also provide the range of values for Good’s coverage across all samples.
P9,Table 5: It appears that virtually all of the sequences for the phylum Synergistetes lie within the genus Fretibacterium. Is this true?
P10, Section 3.5: It appears that urine pH did not decreases for the CON treatment at 13-18 d, but that it did decrease greatly, rather than slightly, in the LD treatment during this period.
P11, Fog.6: The reviewer is uncomfortable with the linear regression analysis of urine pH vs, DCAD level. It seems instead that one could interpret the relationship as a saturation relationship, in which urine pH is maintained as long as DCAD level is above 150, but below this value urine pH declines.
P12, para.2, L6-8: The authors’ logic is not clear here. First of all, isn’t the purpose of the NaHCO3 addition to alter the level of DCAD, rather than as buffer to gastric acidity? Secondly, how can sodium bicarbonate be “essential for stomach health” when NOT adding it had no effect on any measurable rumen variable?
P2, para.2, L11-13: The importance of Prevotella to the rumen is hardly novel. There are many references in the literature identify Prevotella as the most abundant bacterial genus in the rumen. The authors indicate the importance of this genus in protein degradation, but in fact Prevotella are well-known generalists that ferment starches, hemicelluloses, sugars, etc., and their wide use of carbohydrates (always the most abundant components of rations) is probably the major reason they are so abundant.
P12, para.2, L13-19: These are rather meaningless statements. Don’t most ruminal bacterium “play an important role in the process of substance metabolism in rumen”? What does obesity have to with any of this? And how can the elevated abundance of Firmicutes in HD and LD treatments be ascribed to “application of a high forage diet”, when all diets, including CON, had essentially the same forage content (according to Table 1)?
Tables 2-6: For all of these data table, the authors provided pool SEMs and indicate significant differences by superscripted letters, to there is no reason to include standard errors (+/-) for individual mean values.
Minor edits:
Table 2: Change “Total VFA” to “Sum of Ac+Pro+But”. Also, the concentrations are presented at an unreasonable level of precision. They should be rounded to 0.1 mM instead.
Table 3: It would be useful to add an additional line indicating the sum percentage of the four species.
P6, Section 2, L3: Insert “that” before “consumed”; change “and” to “or”.
P.8, para.1: Change “were observed” to “displayed”.
P8, Fig.3 legend, L6: Change “was” to “is”.
P8, para.2, L1: Change “tested” to “detected”.
P8, para.3, L3: Ruminococcus is listed twice.
P9, L1: Capitalize “prevotella”.
P12, para.1, L8: Change “consumed” to “consuming”.
P12, para.1, L12: Include units.
P12, para.1, L18: Change “was” to “were”.
P12, para.1, L19: Change “rumen populations for” to “ruminal abundance of”.
P12. Para.2, L2: Change “unaffected” to “lack of treatment effects on “.
P13, last line: Change “harmness” to “harm”.
Reviewer 2 Report
The article addresses an interesting topic about the plasma calcium level in female goats. The document is well structured and quite easy to follow. I have a few suggestions for the authors:
In all the document authors should write Firmicutes without italic.
Simple summary: In this section, according to the instructions for authors “No references are cited and no abbreviations”.
Line 28: DCAD should be defined, it is the first time that appears in the abstract.
Line 41: ACE should be defined, it is the first time that appears in the abstract.
Introduction: The authors present the subject with a good introduction which provides sufficient background, using enough relevant references.
Line 59: Delete “of” in “… (Cl− and S2−) of per kg of DM….”
Materials and methods: This section should be better explained.
It is an experiment with animals, so an ethical approval is compulsory. (“Interventionary studies involving animals or humans, and other studies require ethical approval must list the authority that provided approval and the corresponding ethical approval code.”). In the document this part is uncomplete. Authors should write the authority that provided approval and the corresponding ethical approval code.
In which productive status was the goats?
Line 93: TMR should be defined, it is the first time that appears in the abstract.
Where the goats allocated in the metabolic cages, or were they only fed there? In this case, were they in the cages the 45 days? This point is not so clear.
Table 1: The units of DCAD in the first row should be deleted. The units are in the last row. DCAD should be defined, tables should be self-explanatory.
Line 129: Authors state that the rumen fluid was acidified to determine VFA. When was it acidified and how? This should be better explained.
Line 137: Authors should explain if the rumen fluid used for the DNA extraction was an aliquot of the fluid collected and it was not acidified. In the text it is not clear.
Line 172: …Tukey…
Authors should write the number of sequences, obtained in the sequencer, before and after filtering them.
With which programme were the alpha and beta diversity calculated and the rest of the microbiota plots constructed?
To analyse the effect of the DCAD levels on the different OTUs the Kruskal-Wallis test recommended, rather than a mixed model. If possible, authors should perform this test.
Results:
Lines 203-205: Rephrase. R. flavefaciens did not increase.
Line 218: …OTU…?
Line 227: Authors should define Ace because it is the first time that appears in the document, excluding simple summary and abstract.
Line 237: Both “principal coordinate analysis (PCoA) based on” Unifrac “distance”… and unweighted “distance”…
Line 266: It would be more explanatory to write here: “that are genera with more than 0.1% of relative abundance.
Table 5: In the tittle: … cation-anion …Phyla should not be in italic. Authors should define letters a, b.
Table 7: Authors should define the parameters. Tables should be self-explanatory. Authors should define letters a, b.
Discussion:
A short introductory paragraph to remember the experiment, after several pages of results, should be appreciated.
Line 350: Authors should check English.
Lines 353-355: Authors should rephrase and introduce a reference.
Line 380-381: … a level too low… ?
References:
Line 522: reference 42: 1991c?
Reviewer 3 Report
Review of Manuscript Animals-1060283
The paper aimed to evaluate the effect of dietary DCAD levels on ruminal fermentation parameters, ruminal microbial population and Ca plasma level. The results are not really surprising, but at least this study confirms that DCAD has no effect on ruminal fermentation and microbiota and confirms the expected effect on plasma Ca and urinary pH. I recommend the authors to consider the following comments and remarks to improve the quality of this paper:
Major comments
The introduction explains well the background and importance of DCAD for dairy cows. The introduction is based on literature (reference 12-18) about the effect of DCAD on dairy cows. This study was done on goats. Therefore, I would recommend to emphasize the introduction on literature from goats or clarify that similar effect are also expected for goats. I think, authors were not really able to justify the realization of the study. Authors expected no effect of negative DCAD. Therefore, I just wonder what the purpose of running such an experiment was.
M&M is described with sufficient detail and results are mostly clearly presented and described with few exceptions (see minor comments). Additionally, in some cases, authors interpreted wrongly some “statistical differences” (se also minor comments)
Discussion is also in general clear
Minor comments
L63: For cows in dry or lactating period? Specify. The study was done in goats. Therefore, I would recommend
L77-79: If authors expected no effect of negative DCAB, why was then this study performed? What will gain the scientific community?
L88-89: Please provide the names of the respective institutions and the number of the approval.
L90: Were animals in dry or lactating phase? Specify
L92-93: Normally, DCAD is given in miliequivalets/kg DM. I would recommend to do it like that for easy comparison between publications.
L104: Measured or calculated?
L106: In Table 1, delete “DCAD (mmol/kg DM) from above of the Table
L204-205: Was this statistically proved? Otherwise, the statements is not valid. Check
L206: Table 3 is somehow confusing. % is presented twice in the table and some values are presented as coefficients. Explain these values/units in the footnotes
L223: In Figure 2, it is difficult to allocate each of the colours in the lines of the graphic. Beside colors, use different type of lines for the 3 treatments. Or it is possible to present one line (average) for each treatment?
L234: In table 4, p-value for Shannon parameter was <0.05. Therefore provide the superscripts i.e. a,b
L293: In Table 6 I think something is wrong with the feed conversion ratio. Values are too high. Please check
L294: I would recommend to give the results (per period) including statistical results in a table to have a better overview
L297: Was this statement based on statistical analysis? Otherwise not valid
L303: Provide the exact p-value obtained
L305: In figure 5, reduce the numbers in the x axis. Give intervals of 5 days
L308: I would recommend also to check a quadratic equation. I think this will fit better
L312: According to the table, the p-value is < 0.01. Check
L314: Use the p-value observed, therefore, p≥0.37
L315: In table 7 check the number of decimals. For example, for variable GSH-Px, 2 decimals is not necessary. Check also other tables
L355: If statistically different, the p-values have to be provided in Table or somewhere else